# Óscar Romero, Ecclesiology, and the Church: Nourished by the Preached Word

Benjamin A. Roberts 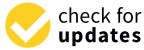

Our Lady of Lourdes Catholic Church, 725 Deese Street, Monroe, NC 28112, USA;
fatherbenjamin16670@gmail.com

**Abstract:** Preaching provides a nourishment that both satisfies and creates hunger. The church is a place of preaching, as well as a subject, an object, and a recipient of preaching. In the multidimensional ecclesial–homiletical relationship, proclamation affirms and enhances ecclesial identity, ponders and interprets the received word of the sacred scriptures, offers challenge and consolation, inspires missionary and cultural extension, celebrates holiness, and proclaims temporal and eschatological hope. These six characteristics offer a lens for homiletical exploration and evaluation. The sermons of Óscar Romero, the martyred Archbishop of San Salvador, provided critical nourishment for the people of his country and beyond. This article provides a brief overview of the biographical, pastoral, and theological details of Romero's life. It then places the six characteristics of the ecclesial–homiletical relationship as a pulpit canopy over a selection of his sermons, revealing the abundant homiletical feast for the church. The preaching ministry of this shepherd nourished his flock through effective and creative engagement with scriptural, magisterial, theological, political, and cultural sources. Óscar Romero shines as an exemplar of homiletical proclamation for ecclesial nourishment.

**Keywords:** preaching; ecclesiology; Óscar Romero; homiletics; liberation; the Transfiguration



## 1. Introduction

*We must not simply read the scroll; we have to eat it.*

—Jennifer Newsome Martin (2021, p. 32)

A wise matriarch from a congregation in Greensboro, North Carolina, once described good preaching as "a meal, and a snack for later."[1] She recognized preaching as a source of spiritual food that sustained her and facilitated her growth, not simply in the moment, but over time. She was nourished by the preached word.

The church, too, is nourished by the preached word. Between the church and preaching, a multidimensional relationship abides. The church is a place of preaching, both temporally and existentially. The church, as the subject of preaching, proclaims the Gospel and the implications of the Gospel from generation to generation. As the object of preaching, the church is proclaimed as the people of God, as the Body of Christ (Rom 12:4–8; 1 Cor 12:12–27; Eph 4:11–16; Col 1:18, 24), as a pilgrim journeying toward the kingdom of God, and as the Bride of Christ (2 Cor 11:2; Rev 19:7–9), among many other metaphors. The church is the recipient, the beneficiary, of preaching who receives a nourishment that, paradoxically, both satisfies and creates hunger. In this essay, focusing on the church as the recipient of preaching, I propose six characteristics of an ecclesial–homiletical lens through which the nourishing power of preaching for the church can be examined. I will utilize these six characteristics to construct a pulpit canopy to amplify the ecclesial nourishment provided by the preaching of Óscar Romero, the martyred Archbishop of San Salvador.

## 2. The Ecclesial–Homiletical Lens

To construct this pulpit canopy and vantage point for observation and exploration, I propose six characteristics for an ecclesial–homiletical lens. These characteristics are

the fruit of the pastoral experience of preaching and the academic experience of teaching ecclesiology. The boundaries of these characteristics are not fixed and immovable walls made of stone, but are more like buoys on lines of rope in a lake marking off one area of swimming or fishing from another, while allowing fluid and free movement across the marked boundary. The use of this lens will focalize insights into how the church is nourished by preaching. The six characteristics are described as follows.

The first characteristic of the ecclesial–homiletical lens seeks words, themes, and concepts that affirm and enhance ecclesial identity. These include metaphors to describe the church, as well as aspects of the church's life, foundational mission, virtues, wounds, and depth of union with Christ. It also includes the prophetic call from within the church to the church for greater fidelity. When this characteristic is present, the hearers of the sermon come to understand the nature of the church and of corporate and individual union with it, and they are encouraged to experience and value communion with one another and with the saints. The first characteristic strengthens identity.

The second characteristic looks for the ways through which the preacher ponders and interprets the received word of the sacred scripture. This characteristic observes the methods by which the scriptural texts, and by extension liturgical and ecclesial texts, are engaged in the sermon. When this aspect is present, the hearers are invited into the sacred texts to find and receive nourishment. The second characteristic affirms scriptural foundations.

The third characteristic notes how the preacher offers challenge and consolation. The challenge can be directed to individuals or groups within the church, to the broader society or government, and to the enemies of the church. The consolation is generally directed toward the faithful. When this characteristic is present, the hearers are invited to conversion and greater fidelity.

The fourth characteristic of the ecclesial–homiletical lens observes inspiration for missionary and cultural extension. This concerns the engagement of the church and the members of the church with the wider culture, with society, and with non-believers and those of other faith traditions. If the first two characteristics can be seen as drawing the attention of hearers inward toward the church's core identity and resources for strengthening faith, this characteristic looks for the outward engagement encouraged by the preacher.

The fifth characteristic notes the ways in which the preacher celebrates holiness. This is generally achieved by pointing to a particular figure or groups who provide witness and whose witness bears light and encouragement for the preacher and the hearers of the preaching. When this characteristic is present, the hearers of the sermon are offered an example to honor and imitate the exemplary figure(s) in ways that are appropriate to their lives.

The sixth and final characteristic of the ecclesial–homiletical lens notes the proclamation of temporal and eschatological hope. This characteristic looks forward not only to ultimate salvation and liberation, but to a greater manifestation of justice and fidelity in the present and future. This characteristic provides the motivation for missionary and cultural extension and enlightens all of the other characteristics. When this characteristic is present, the hearers experience hope.

These six characteristics of the ecclesial–homiletical lens facilitate the examination of homilies for ecclesiological doctrine and content. The homiletician can utilize this lens to explore how the church is nourished through preaching. The preacher could utilize this lens in the construction of sermons in order to highlight ecclesiological themes as might be appropriate on certain feasts, or during the season of Easter when passages from the Acts of the Apostles are included in the lectionary. This lens might also be of use to the systematic theologian seeking to harvest the seeds and fruits of ecclesiological insights from the homilies of a preacher or preachers. Therefore, the characteristics of this ecclesial–homiletical lens might serve as constructive for the preacher, evaluative for the homiletician, and generative for the theologian.

The following section offers a biographical and theological sketch of Archbishop Óscar Romero and provides an overview of the sermons examined in this essay.

### 3. Overview of the Preacher and the Sermons

Óscar Romero lived an ecclesial life.[2] He entered seminary in his teens, studied theology in Rome, and was ordained a priest on 4 April 1942 (Brockman 2005, pp. 35–38). He was appointed Archbishop of San Salvador in February 1977, and was assassinated at the altar on 24 March 1980 (Brockman 2005, pp. 4, 245). The lifeblood that flowed through his years of pastoral ministry continues to pulse in the recordings of the radio broadcasts and texts of his homilies during his three short years as Archbishop (Colón-Emeric 2018, pp. 47–48).

El Salvador in the 20th century was marked by profound geopolitical volatility. Elite landowning families exploited their Indigenous workers with impunity. Repressive governmental forces tortured and executed thousands of Salvadorans perceived to pose a threat to whichever military dictatorship held sway; targets included Roman Catholic priests, teachers, and labor organizers (Mong 2021, pp. 41–43). The United States funded several Salvadoran groups that, it was learned, were functioning as right-wing death squads (Gandolfo 2013, pp. 64–69). Archbishop Romero interceded more than once, seeking to halt the flow of resources from the U.S. to these groups. From the time of Romero's priestly ordination in 1942 to his murder in 1980, he was preaching in communities terrorized by militarized violence and riven by the fear of brutal governmental reprisals against dissent (Mong 2021, pp. 44–48).

Michael Connors, CSC, suggests that Romero is a "hero of the spoken word, a homiletic patron saint for our times" (Connors 2015, p. 93). Alma Tinoco Ruiz summarizes his preaching in this way: "Romero effectively listened to the voice of God in both Scripture and the concrete circumstances of the poor, oppressed, and marginalized people and let them illuminate each other. As a result, his homilies are both prophetic and pastoral. He spoke for God and on behalf of the people" (Tinoco Ruiz 2020, p. 5).

The preaching of Óscar Romero radiated his theological perspective.[3] Edgardo Colón-Emeric, in his magisterial work on Romero, notes that the Transfiguration was the "focus of his theological vision" (Colón-Emeric 2018, p. 22). Margaret Pfiel elaborates: "His Transfiguration homilies and related pastoral letter bear witness to his commitment to plumbing the depths of the feast's theological meaning as he exercised his episcopal role as preacher and teacher" (Pfiel 2015, p. 66).

I have chosen the six sermons that focus on the Transfiguration for analysis with the ecclesial–homiletical lens. Three were preached on 6 August and three were preached on the second Sunday of Lent.[4]

1. 6 August 1977, Feast of the Transfiguration, *The Church, The Body of Christ in History* (Romero 1977).
2. 19 February 1978, second Sunday of Lent, *The Church, A Spiritual Israel* (Romero 1978a).
3. 6 August 1978, Feast of the Transfiguration, *The Son of Man, Light of the Pilgrim People on Earth* (Romero 1978b).
4. 11 March 1979, second Sunday of Lent, *The Transfiguration of the People of God* (Romero 1979a).
5. 6 August 1979, Feast of the Transfiguration, *The Church's Mission In the Midst of the Nation's Crisis* (Romero 1979b).
6. 2 March 1980, second Sunday of Lent, *Lent as God's Plan for Transfiguring the Peoples Through Christ* (Romero 1980).

### 4. Characteristics of the Ecclesial–Homiletical Lens

In what follows, I employ the six characteristics of the ecclesial–homiletical lens to examine the homiletical nourishment offered by Archbishop Óscar Romero in his Transfiguration homilies.

*4.1. The Preacher Affirms and Enhances Ecclesial Identity*

The preached word affirms and enhances ecclesial identity. This first characteristic of the ecclesial–homiletical lens looks and listens for words, themes, metaphors, and examples

that emphasize the church's nature, life, mission, virtues, wounds, and union with Christ. When we utilize this lens to examine the Transfiguration homilies of Romero, several themes emerge. This section provides examples from the selected homilies that foreground ecclesial identity.

In the homily of 6 August 1977, entitled *The Church, The Body of Christ in History*, Romero offers: "To understand the meaning of all this we need only to look at our own people. I want to tell you, dear Catholics, that all of us here as church are the transfiguration of Christ. We are a people enlightened by faith, encouraged by great hope, and united by great love" (1.1). In the same homily, he continues:

> This is the church of hope. The church has inspired great hope in our hearts precisely because she no longer finds her power in worldly realities and because she is now lacking the support that people offer her out of self-interest. She has learned to be free of all that in order to be faithful to the Gospel. Now in her poverty the church knows that she is with the poor, and all those who want to live with her and share her hopes must find support in the weakness of the derided Christ, in the weakness of the church as spouse of Christ, in her poverty, in her Gospel, and in her authentic following of the Lord. (1.6)

In the first homily, preached on 6 August, which is the liturgical feast of the Transfiguration, Romero highlights the glory or Transfiguration of the church facilitated through her poverty and standing with the marginalized. In the second homily, preached on 19 February 1978, the second Sunday of Lent, and entitled *The Church: A Spiritual Israel*, he proclaims that the suffering of the cross is an authenticating sign of the church and an iconic symbol of resurrection:

> All the life and all the history of Christianity are moving toward the cross and toward the resurrection. That is why we should not be surprised, sister and brothers, that the church must bear many crosses, for otherwise she would not have much of a resurrection. A church that seeks accommodation and prestige without the pain of the cross is not the authentic church of Jesus Christ. (2.6)

While poverty and suffering point toward attributes of the church's identity, in his third Transfiguration homily, entitled *The Son of Man, Light of the Pilgrim People on Earth* and preached on 6 August 1978, Romero explores a different metaphor for both ecclesial identity and mission; that of the lamp: "The church feels that she is God's lamp; she is light taken from the glowing face of Christ to illumine the lives of men and women, the lives of nations, the complicated problems that people create in history. The church feels the need to speak and shed light, just as a lamp in the night feels the need to light up the darkness" (3.3).

Continuing this metaphor in the same homily, Romero describes the unique nature of the church's presence and activity: "For us more than anyone else the word of Christ becomes a command so that we truly become a church that shines like a lamp in the night, a church that is not confused with other lights but always gives forth the pure light of Christ. The church, sisters and brothers, reveals the transfigured Christ" (3.6).

Moving from lamp to shining light, in his fourth Transfiguration homily, preached on the second Sunday of Lent, 11 March 1979, and entitled, *The Transfiguration of the People of God*, Romero expounds on the implications of the church as a shining light:

> The church is meant to be a steady light that shines forth to serve and illuminate the world. Accordingly, we must shed light on the realities around us. Don't take it amiss, therefore, if after describing the shapes and forms of the church and meditating on the Gospel that helps us build up the church, we direct our sight toward what is around us in order to affirm what is good and to denounce and reject the bad and sinful things that are taking place. (4.12)

In his fifth Transfiguration homily, preached on 6 August 6 1979, and entitled, *The Church's Mission In the Midst of the Nation's Crisis*, the Archbishop speaks of the duty of the church to contribute to society. Romero notes that this contribution comes from the very

nature of the church's identity. He proclaims: "On this day the church is contributing what it is her duty to contribute: all the richness of the church, all the maturity of the diocese, all the convictions of priests and bishop and people. That is to say, from our identity as church we are expounding what we think and what we can offer our county at this critical time" (5.2).

Archbishop Romero preached his final Transfiguration homily on 2 March 1980, the second Sunday of Lent. This was only 22 days before he was assassinated. In a homily entitled, *Lent as God's Plan for Transfiguring the Peoples Through Christ*, he speaks boldly of the church's concern for the human person. Echoing his light metaphor with the guidance of a star, he preaches:

> The church must always keep the human person in sight. This is the star that guides her along the way. She is often misunderstood and often maligned because many people want to make their earthly projects prevail. All the church is concerned about is the human being, the child of God, and that is why she grieves over all human corpses being found, over the torture of human beings, over the suffering of human beings. For the church the goal of all projects must be this great project of God: the human being. Every person is a child of God, and in every person killed the church venerates a sacrificed Christ. (6.5)

While in earlier homilies Romero focuses on the glory revealed in the Transfiguration, the final two homilies attend to the humanity of the church and how it shares in the suffering of Christ.

In his Transfiguration homilies, Óscar Romero affirms and enhances ecclesial identity for his hearers through exhortation, metaphor, and the defense of the human person. Utilizing the biblical imagery of the Transfiguration, the cross, the lamp, and the shining light, he offers encouragement and affirmation to his flock. With a great sense of ecclesial identity, his hearers would have been prepared to ponder the revealed word of God in Romero's focal scriptural texts, which will be explored in the next section of this essay.

### 4.2. The Preacher Ponders and Interprets the Received Word of the Sacred Scriptures

The second characteristic of the ecclesial–homiletical lens explores how the preacher ponders and interprets the sacred scriptures. In addition to commenting on the scriptural texts and utilizing them for illustrations, such exploration includes how the preacher approaches and explains the task of interpreting the texts.

Romero engages the scriptural texts and the homiletical task through the use of what the Pontifical Biblical Commission will later call "actualization." Actualization is described as "sincerely seeking to discover what the text has to say at the present time" (Pontifical Biblical Commission 1993, IV.A.1). Romero actualizes the scriptural texts for his hearers with careful exegesis, drawing out implications and contemporary applications.

In his first Transfiguration homily, Romero invokes the scriptural text and invites the hearers to stand within that text:

> Like Saint Paul the church turns toward Christ and asks, 'Who are you?'. The church asks Christ, 'Who are you so that I might follow you, so that I might lend you my feet to walk on the paths of my country's history, so that I might lend you my mouth to proclaim your message and my hands to work and bring about your kingdom?' (1.3)

Standing physically and existentially within the church, the hearers are offered an entry into the living word of the scriptures. In his second homily, Romero proposes that preaching is an echo of the voice of Christ. He also notes the contextual framework of the events of the week; this is a key feature of his homiletical method. He announces:

> The one who preaches in the cathedral, as in all church pulpits, is nothing but a humble echo of the divine voice that guides us, Christ the teacher. The one who preaches does nothing more than take this eternal word and illuminate with it

the realities of our journey through history. That is why I take care every Sunday
to place the word of God within the framework of each week's history. (2.1)

In the same homily, he ponders the call of Abraham and the freely given divine call to
all believers, which forms a people with a promise and a mission: "God takes the initiative
in forming a people and extends to them his hopes and promises. This is the great mission
of Abraham: 'I will make of you a great nation,' a people from whom the Redeemer will be
born" (2.5).

Concluding the second homily, he proclaims the mystagogical truth that the word
proclaimed is the word celebrated and ritualized in the celebration of the Eucharist. He
invites the assembly to union with Christ through nourishment from the word and the
altar: "Dear sisters and brothers, this is the Liturgy of the Word over which the transfigured
Christ has presided today here in our cathedral. Christ is now not only word but becomes
host and chalice; he becomes communion and life. As we take Communion, let us identify
ourselves with his thoughts. Let us live our Eucharist" (2.8).

Romero guides his assembly to recognize and understand the presence of Christ in
the word proclaimed within the liturgy celebrated. It is Christ who presides over the
proclamation from the pulpit and at the altar. In his third Transfiguration homily, he invites
his hearers to heed the voice of the God the Father in the scriptural text:

In commenting on God's word as we celebrate our country's most glorious feast
today, sisters and brothers, I find in the Son of Man and in the splendor of his
glory the light that illumines this pilgrim people on earth. That's why we hear—
with all the logic of a God who knows better than we who this transfigured One
is—the command that each one of us should take away as the message of this
mystery, 'Listen to him.' (3.1)

In a passage from his fourth Transfiguration homily, Romero reveals the title he has
given to this preaching event, a feature that he often included in his homilies:

The title, therefore, that I'm going to put on my homily this morning is this: 'Lent,
the Transfiguration of the People of God.' I will develop this reflection in accord
with the three readings which suggest to me these three ideas: first, the covenant
with Abraham that gave rise to the people of God; second, the transfigured
Christ as the model and the cause of the transfiguration of our people; and third,
the need for us, as God's people, to be transfigured here and now so that the
Gospel on which we meditate today—and which I've always tried to promote—
becomes a living word that speaks to me, to you, to our families, and to our
communities. (4.2)

Romero references the work of biblical scholars to provide a historical context for
his hearers. He demonstrates the value of academic biblical studies for the preacher and
the members of the assembly who desire to hear the living word of God. In his fifth
Transfiguration homily, he teaches: "In the first reading Daniel saw the figure of a man
surrounded by the glory of God. Scripture scholars say that this figure is the glorified Christ,
surrounded by all who are saved. This is the transfiguration we long for: a church that will
be glorified but never loses sight of her exalted dignity while still on pilgrimage" (5.8).

In his final Transfiguration homily, Romero emphasizes that his exploration of the
scriptural text and his preaching are not ends in themselves but in service of an encounter
with Christ. He proclaims: "My words have no other aim than to be a humble echo of this
word of God that becomes incarnate in Christ as a guiding light for all peoples. He is the
one whom we most urgently need to listen to, as God commanded us" (6.1).

In the same homily, he provides a brief overview of the entire scriptural message
through the lens of the Transfiguration. He offers to his hearers an analysis of the scriptural
text that is historical, theological, and contextual:

In today's gospel we see two outstanding figures of the Old Testament, Moses
and Elijah, the great legislator of the people and the great prophet of the people.
We see also this great truth that we are trying to understand: that the transfigured

Christ, standing between Moses and Elijah, is the fulfillment of the whole of Israel's history. Moses and Elijah, the patriarchs and the prophets—that whole braid of gold that God was weaving in the history of Israel—had one objective: to bring us the Redeemer, to have the Son of God made man born of that race. (6.4)

The Transfiguration homilies of Archbishop Romero demonstrate a pastoral, historical, and contextual engagement with sacred scripture. Romero utilizes the scriptural text as a lamp by the light of which the circumstances of his day could be read. Through his exploration of the texts, he also highlights the task and method of the preacher. In so doing, Romero implicitly offers to his hearers a way to ponder and interpret the scriptures for themselves in their homes and communities.

The first two characteristics of the ecclesial–homiletical lens affirm identity and ponder the sacred text. Building upon these two characteristics, the next section of this essay will explore how Romero offered both challenge and consolation through preaching.

### 4.3. The Preacher Offers Challenge and Consolation

The third characteristic of the ecclesial–homiletical lens explores how the preacher invites conversion and offers healing. The words of challenge and consolation can be directed to the members of the assembly, to particular groups, and to society at large. In his Transfiguration homilies, Archbishop Romero offers direct challenges to the powerful and hopeful consolation to those who are suffering.

Romero issued pastoral letters, teaching documents published by a bishop to his diocese, on 6 August 1977, 1978, and 1979. He introduces his pastoral letters in his homilies on each of those occasions.[5]

Presenting his second pastoral letter in his first Transfiguration homily, Romero offers his intention in the letter, viz., "so that doubts are dispelled and so that all who have adhered unconditionally to the pastoral lines of the archdiocese may feel more confident that we are walking in the ways of Jesus" (1.1). He continues that this letter is "also for those who still have some reservations, those who love the Church but still wonder whether the bishop has become a communist and whether the priests are preaching subversion and violence" (1.1). Romero concludes his references to the letter's intended audiences by proclaiming that it is also "for those who hate the church and malign her, so that they will know that they are maligning the Body of Christ and will be converted." (1.1) This is a very dangerous statement for Romero to make, because to address his adversaries directly, to speak challenge to the powerful, is to invite suffering for himself and for his people.

In the same homily, Romero notes that suffering is a consequence of the church's fidelity to the dignity of the human person: "The church preaches human development, and for preaching this promotion of people, for waking people out of their unhealthy state of conformity, and for urging them to be active in their own destiny, the church must suffer" (1.4).

In his second Transfiguration homily, preached on the second Sunday of Lent in 1978, he utilizes the metaphor of light to promote the dignity of personal and communal history:

Each of you, dear brothers and sisters, has your own unique history, your family history, and your community history. It would be impossible to describe here those concrete histories; this is the work to be done by each person. Each of us must allow the Gospel to shed light on our hopes, our plans, our disillusionments, our failures. We need the light of the word of God so that we may always live with faith and hope. (2.1)

He continues to shine the metaphor of light in his third Transfiguration homily as he offers words of consolation and hope to the faithful: "This is a nighttime of our history; it is the dark journey of our time. These difficult hours that our land is passing through seem like an endless night—until the sun of the transfiguration becomes daylight, inspiring hope in the Christian people and lighting up our path. Let us follow it faithfully" (3.3).

In his fourth Transfiguration homily, Romero offers words of challenge to those who suggested that his preaching and pastoral activity were more subversive political activities

than manifestations of theological and pastoral mandates. He asserts: "Have no doubt, if our archdiocese has given rise to conflicts, it's because of our desire to be faithful to this new evangelization that is demanded by the Second Vatican Council and the meetings of the Latin American bishops. This evangelization must be fearless and unswerving, and that's why we prayed earnestly to the Virgin of Peace" (4.2).

In his fifth Transfiguration homily, preached on the feast day of the city of San Salvador, which means *Holy Savior* in Spanish, Romero proclaims the dignity of the people as beloved of God and the responsibility to listen to the word of the Lord. He preaches: "Every year on August 6 we hear, as we heard today in our church's liturgy, the voice of the Father proclaiming that our patron is the same Son in whom he takes delight and that our duty as his people is to listen to him. That constitutes our most precious historical and religious legacy and gives rise to our greatest hopes as Salvadorans" (5.1).

In his final Transfiguration homily, preached only three weeks before his assassination, Romero announces the unity of redemption, liberation, and salvation, specifically within the liturgical context of Lent and the historical context of his country. He proclaims:

> We are celebrating redemption, which means the same as liberation and salvation, and liberation is precisely what our people need. Our preparation for Holy Week and Easter—when we celebrate the mystery of human redemption—is so profoundly inserted into the history of our Salvadoran people that we can truly say that Lent and Holy Week are made for us. They are the celebration of our own redemption. (6.1)

In his Transfiguration homilies, Archbishop Romero offers both words of healing and critical calls to conversion. His words highlight the historical context for the hearers, reflecting his theological commitment to the action of God in the concrete experiences of the faithful. Romero announces consolation rooted in the dignity of the person, the mission of the church, and the present activity of the Lord. In the next section, the encouragement and consolation offered to the faithful will be explored through missionary and cultural extension.

*4.4. The Preacher Inspires Missionary and Cultural Extension*

The fourth characteristic of the ecclesial–homiletical lens concerns the engagement of the church and the members of the church with the wider culture, with society, and with non-believers and those of other faith traditions. This characteristic attends to the radiation of the church's mission and ministry into the wider culture and society.

In his first Transfiguration homily, Archbishop Romero announces the impact in the present moment of the future kingdom proclaimed by the church. The mission of evangelization is proclaimed by a community of love and unity. He preaches: "The message of the approaching kingdom of God is the message the church continues to preach. The kingdom of God draws near, and when people understand this message of twenty centuries ago, proclaimed now by evangelizers in 1977, they love one another, they create community, and they detest differences"(1.3).

In the second Transfiguration homily, he seeks to inspire the community of evangelizers, that is, the whole people of God, to have confident trust in God, even regarding their health. He encourages them:

> Being in good health is not as important as trusting in God. This message is for all of you who preach, all you who proclaim God's word, all you who bring the community together as church, all you who teach the true meaning of the Gospel in Christian schools, and all you who want to live in family as true Christians—do not trust in yourselves, but trust in God. (2.7)

Romero draws some careful distinctions in his third Transfiguration homily. He clarifies that the mission of the church extends into the secular realm, but is not completely identified with it. The church extends into the culture by shining the light of the gospel on secular conditions:

This is the church's job; without leaving her proper sphere, she undertakes the difficult task of shedding light on our realities. The church defends the right of association, and she promotes the dynamic activity of raising consciousness and organizing the common people to bring about peace and justice. From her vision of the Gospel, the church supports the same just objectives that the people's organizations seek, but she also denounces the injustices and the acts of violence committed by the organizations. That's why the church cannot be identified with any organization, not even with those that call themselves and feel themselves to be Christian. (3.3)

In the same homily, he provides an important challenge to those who exercise leadership in all strata of society. He places the role of secular leadership within the context of a Christian vocation, thereby extending the mission of the church through the members of the church. He preaches:

I address all those who have achieved any degree of leadership among the people, those who by their professions or their organizational skills hold important posts, and all those who are called leaders even if only within a modest domain. Sisters and brothers, in the name of Christ, help to clarify the reality; search for solutions; don't shirk your vocation as leaders. Realize that what you have received from God is not to be hidden away in the comfort and welfare of a family. Today more than ever the nation needs your intelligence. (3.5–6)

In the fourth Transfiguration homily, Romero speaks prophetically regarding care for creation:

You know yourselves that the air and water and everything we humans touch are contaminated, Even though we keep polluting nature more and more, we still need it to live. We fail to understand that we have a commitment with God to care for nature. We cut down trees, and we waste water even when there is such a water shortage. We don't care about the foul-smelling buses that are poisoning our environment. We don't care about where garbage is being burned. All these matters have to do with our covenant with God, and we must be mindful of the consequences since the population density in El Salvador is very high. Dear fellow Salvadorans, let us care for nature out of a truly religious sense so that it doesn't continue to become impoverished and die. This is our commitment to God, who is asking us for our collaboration. (4.3)

In his fifth Transfiguration homily, Romero announces the mandate within the church's mission to name and challenge injustice. He boldly proclaims: "Sisters and brothers, let us be clear about the mission of the church, which is evangelizing and working for justice. That mission should not be confused with subversive campaigns—unless, of course, you want to call the Gospel subversive, since it *does* seriously question the foundations of an order that should not exist because it is unjust!" (5.4–5; emphasis original).

In his sixth and final Transfiguration homily, Archbishop Romero places the call for social justice within the liturgical season of Lent. He offers a call to personal conversion that extends into societal transformation:

Lent should awaken in us a sense of social justice. Let us therefore celebrate our Lent in this way, by giving to our sufferings, our bloodshed, and our sorrows the same value that Christ gave to his condition of poverty, oppression, abandonment, and injustice, and thus transforming it all into the saving cross that redeems the world and all people. Let us also, with hatred for no one, be converted so that we can give spiritual and material aid, despite our own poverty, to those who are perhaps even more needy than we. (6.2)

In his Transfiguration homilies, Archbishop Romero nourishes the church by inspiring missionary and cultural extension. This characteristic observes how the homily continues in the lives of the faithful. Prophetically, Romero offers his hearers the homiletical sustenance

to speak truth to power. In the next section, which analyzes the celebration of holiness, we will see him elaborate on images of the perfection of Christian power.

*4.5. The Preacher Celebrates Holiness*

The fifth characteristic of the ecclesial–homiletical lens notes the ways in which the preacher celebrates holiness. This is often achieved by pointing to a particular figure or to groups who bear witness. This witness brings light and encouragement for both the preacher and community of listeners. It would also include exercises of spiritual practice and devotion.

In his second Transfiguration homily, Romero celebrates holiness in actions characterized by faith, pointing to Abraham, Paul, Timothy, and to his hearers themselves. He is careful to note that holiness begins with the invitation of God and involves a cooperative response.

> I am happy, sisters and brothers, that the Israel created by Abraham through his act of faith is prolonged in the people of God, reaching as far as yourselves, authentic Christians living in 1978 and reflecting on this word of God. This is the same saving will of God who wants to save all, as Paul tells Timothy. He wants to sanctify us and the initiative is his. (2.7)

In his fourth and fifth Transfiguration homilies, Archbishop Romero celebrates the holiness of the local martyrs, including Rutilio Grande, who had been assassinated in March 1977, and Alirio Napoleón Macías, who was assassinated in August 1979. Fr. Rutilio Grande, SJ, had been a close friend of Romero for many years.[6] Referencing a pilgrimage from Aguilares to El Paisnal beginning that morning, he preaches, "This pilgrimage of silence, prayer, and reparation is not leaving Aguilares for El Paisnal, and it should be clear that it is a prayerful march whose object is to honor the memory of Father Rutilio Grande on the second anniversary of his murder." He continues, "Thanks to the Gospel message that Father Grande left in Aguilares, the church there is marked with the seal of authenticity." (4.1)

At the beginning of his fifth Transfiguration homily, Romero preaches:

> Among all these priests who have come to express their profound sense of ecclesial communion, I want us to feel also the presence of a much loved priest who died prematurely at the hands of assassins. Father Alirio Napoleón Macías is laid out now there in his parish of San Esteban Catarina, but he is present with us now as he was on so many other occasions. (5.1)

Romero points to the holiness of living faith and the holiness of local martyrs in living memory. In his final Transfiguration homily, he references his radio audience in Costa Rica and explains his normal preaching method to them. In this homily he celebrates the holiness of his flock who carried the light of the gospel into the events of each week, thereby building up the church. He preaches:

> We will now try to incarnate into our archdiocesan church these themes of Lent and the transfigured Christ on which we have been reflecting. So for those who are not used to hearing our homilies, I explain that we present here a kind of chronicle of the week. We recount the work that is being done in the church, and we do so not out of vanity but out of a desire to share, among us who believe in Christ and make up the Church, the ideals which we want to embrace more every day in order to form the true church of Jesus Christ. (6.9)

Archbishop Romero celebrates holiness by highlighting faith, martyrdom, and the concrete responses of his community to the invitation of the gospel. Holiness, as proclaimed by Romero, was lived in this life, in the common and tragic realities of each day in his country.

In the final section, we will look at Romero's proclamation of temporal and eschatological hope.

### 4.6. *The Preacher Proclaims Temporal and Eschatological Hope*

The sixth and final characteristic of the ecclesial–homiletical lens gazes ahead. This characteristic looks forward not only to ultimate salvation and liberation but to a greater manifestation of justice and fidelity in the present and future.

In the first Transfiguration homily, Archbishop Romero proposes the hope that the Gospel would bring about the unity of humanity. He proclaims: "The Good News that Jesus brought was the announcement of great hope, the formation of a humanity where all would be sisters and brothers to one another and God would be seen as Father of all" (1.3). He offers a vision of conversion and holiness in which the church would be an agent of sanctification and a sign of the reign of God. He encourages his listeners to continue on with firm resolve:

> Let us persevere in hope, not only the hope that the church will continue working in all her authenticity, beauty, and unity, but also the hope that this church, made more beautiful in persecution, may be understood by the persecutors themselves to be free of hatred and resentment. May the church know how to apply all the rich potential that Christ offers her to sanctifying family life, to sanctifying politics, to sanctifying the economy, and for making it possible for Christ to say also in El Salvador, 'The kingdom of God is near. Be converted.' (1.7)

In the second Transfiguration homily, he invites his hearers and the larger community to hope for a just and free society by following Christ. He explains:

> Through his cross and resurrection Christ is summoning human beings to their true greatness as individuals and as society. There cannot be a new society or a new way of living without Christ. There cannot be well-being for all people without the justice of Christ the Redeemer. He alone can inspire selfish people to repent. He alone can inspire resentful people to work honestly and honorably. He alone can give a true meaning to Christian liberation and redeem us from sin and death so that we can participate in his glory. (2.7)

In the fourth homily, Romero addresses those advocating for the rights of the poor and the oppressed. He offers them hope, confirming that they are seeking the values of the reign of God and affirming the church's understanding of their goals. He proclaims:

> There are many of you fighting for your rights in our land. You are our beloved workers and *campesinos* and members of the people's political organizations. The church cannot identify directly with you, but she understands you well; she also demands the same justice and fairness that you demand because they are a reflection of the kingdom of God, which will gather together for eternity all that is good in the world. (4.8)

In the same homily, he announces the eschatological hope of heaven and the union of the transfigured Christ with the poor and suffering:

> This is the transfiguration of the Christ today. He goes up the mountain not to distance himself from humankind but to give us an example and to tell us that the only thing of value is the heavenly happiness of being a child of God—'This is my beloved Son.' Being poor or being rich is not important; what matters is being a child of God, especially that child in whom God delights. I urge you all to be such a child, brothers and sisters. (4.14)

In his fifth Transfiguration homily, Romero announces the Transfiguration of the people in Christ. The hope that Romero proclaims affirms the dignity of the people as beloved children and signs of God's glory: "Through his transfiguration Christ is telling us that this is our goal: to become new, transfigured men and women, clothed in God, people of whom God can say, 'My beloved child with whom I am well pleased'" (5.8).

In his final Transfiguration homily, Romero invites his hearers to prayer and assures them that the experience of pain and suffering would be transfigured into glory in Christ. He announces:

> Let us pray, dear sisters and brothers, because the situation in our country is very difficult. Nevertheless, the vision of the transfigured Christ during this time of Lent is showing us the path we must follow. The path of our people's transformation is not far off. It is the path that God's word indicates to us today: the path of cross and sacrifice, of blood and pain. But filled with hope, let us keep our sight fixed on the glory of Christ, who is the Son chosen by the Father to save the world. Let us listen to him! (6.17)

In his Transfiguration homilies, Archbishop Romero proclaims hope in this life and the next. He offers his hearers hope based on the incarnation of the gospel value of justice in Salvadoran society. He does not simply point to a heaven in which, one day, things will be better. He offers hope in eternal life, and also hope in this life that, even amid sorrow and sacrifice, the power of Christ would transfigure each of them, their community, and eventually, their country.

### 5. Conclusions

The Transfiguration homilies of Óscar Romero provided critical homiletical nourishment to the faithful of his country in the midst of a protracted national and ecclesial crisis. The ecclesial–homiletical lens provided a method to explore these homilies and harvest some of the aspects of ecclesial nourishment. The fruits of this harvest offer preachers a model for ecclesial focus in homilies, enable homileticians to evaluate ecclesial nourishment, and provide theologians with the seeds to construct an ecclesiology rooted in preaching. The gift of preaching nourishes the preacher, the homiletician, and the theologian.

Preaching nourishes the church. It provides a meal for today and sustenance for the pilgrim road ahead.

**Funding:** This research received no external funding.

**Institutional Review Board Statement:** This research did not require institutional review.

**Informed Consent Statement:** This research did not require informed consent.

**Data Availability Statement:** No new data were created or analyzed in this study. Data sharing is not applicable to this article.

**Conflicts of Interest:** The author declares no conflict of interest.

### Notes

[1]  Deria Foster-Moore said this to me at St. Paul the Apostle Catholic church in the Fall of 2009.

[2]  For additional perspective on Romero's continuing ecclesial influence, see (Francis 2019).

[3]  For insights on Romero's theology of Transfiguration, see (Calleja 2021) and (Kibbe 2019).

[4]  For this essay I have used the English translations of Archbishop Romero's homilies provided on www.romero.org.uk. The homilies are provided in PDF format. The first reference for each homily includes the link to the text. Subsequent references cite the number of the homily (1–6) and the page number of the reference in the PDF text. As an example, a reference 2.6 would refer to the sixth page of the second homily.

[5]  For further reading on the pastoral letters, see (Romero 2020).

[6]  On the relationship between Romero and Grande, see (Kelly 2015, pp. 111–22).

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
