# Peer review of "Óscar Romero, Ecclesiology, and the Church: Nourished by the Preached Word"

_religions, doi:10.3390/rel15030322_

Round 1
Reviewer 1 Report
Comments and Suggestions for Authors
Overall, this article is very well written and I found the six characteristics of the ecclesiastical-homiletic relationship helpful both in composing a homily as well as in analysing homiletic texts. The author of the article is evidently well acquainted with Romero's homilies and with his life and context. However my first major critical comment concerns the methodology and argument of the article. What are the sources for the six characteristics? Why not more or less than six? Or are these merely arbitrary characteristics? Failing to identify a logical argument or methodological framework underpinning these six characteristics risks rendering the article as merely descriptive in nature without a robust argument to support it.
My second concern is that there is lack of theological argument about why any of the characteristics render the homily effective in bringing about social change. For instance, the author can buttress her argument that the preacher celebrates holiness by referring to the exemplarist moral theory of Linda Zagzebski. For the last characteristic that speaks of hope, the author might want to consider using contemporary theories of hope (see, for instance, Margaret Urban Walker) to strengthen her argument further.
Overall, the article is well written but it might benefit from cutting down on the longer quotations to allow for a stronger argumentative component in order to increase its effectiveness and incisiveness.
Author Response
I understand the concerns regarding the length of quotes. However, in order to preserve the content and beauty of Romero's words, I wanted to maintain the more substantial quotes.
Additionally, I appreciate the question about the six characteristics leading to social action. I am not yet convinced that all of the characteristics lead to social action. Since the preacher surrenders the interpretation and impact of the sermon to the listener with the act of preaching, a direct social action would be hard to measure. I am grateful for the suggestion and will look to engage it more fully as this project may expand beyond this initial essay.
Reviewer 2 Report
Comments and Suggestions for Authors
Congratulations on a very fine article. Using the six Transfiguration homilies of Archbishop Romero to demonstrate the ecclesial-homiletical lens is brilliant, given Romero was an excellent preacher. This is a fine contribution to the study of homiletics. I recommend the article be expanded into a book so the author could delve more deeply into the dense content of Archbishop Romero.
Author Response
I am incredible grateful for your generous and insightful comments about my essay. I will certainly consider developing a book project!